# Sensitivity to imidacloprid insecticide varies among some social and solitary bee species of agricultural value

Blair Sampson[1]*, Aleš Gregorc[2], Mohamed Alburaki[3], Christopher Werle[1], Shahid Karim[3], John Adamczyk[1], Patricia Knight[4]

1 USDA-ARS Thad Cochran Southern Horticultural Laboratory, Poplarville, Mississippi, United States of America, 2 Department of Agriculture and Life Sciences, University of Maribor, Maribor, Slovenia, 3 Department of Cell and Molecular Biology, University of Southern Mississippi, Hattiesburg, Mississippi, United States of America, 4 Mississippi State University, Coastal Research and Extension Center, Starkville, Mississippi, United States of America

☯ These authors contributed equally to this work.
* blair.sampson@usda.gov

**Data Availability Statement:** Raw data are available through the AG Data Commons Portal as electronic supplementary material at the following

## Abstract

Pollinator health risks from long-lasting neonicotinoid insecticides like imidacloprid has primarily focused on commercially managed, cavity-nesting bees in the genera *Apis*, *Bombus*, and *Osmia*. We expand these assessments to include 12 species of native and non-native crop pollinators of differing levels of body size, sociality, and floral specialization. Bees were collected throughout 2016 and 2017 from flowering blueberry, squash, pumpkin, sunflower and okra in south Mississippi, USA. Within 30–60 minutes of capture, bees were installed in bioassay cages made from transparent plastic cups and dark amber jars. Bees were fed via dental wicks saturated with 27% (1.25 M) sugar syrup containing a realistic range of sublethal concentrations of imidacloprid (0, 5, 20, or 100 ppb) that are often found in nectar. Bees displayed no visible tremors or convulsions except for a small sweat bee, *Halictus ligatus*, and only at 100ppb syrup. Imidacloprid shortened the captive longevities of the solitary bees. Tolerant bee species lived ~10 to 12 days in the bioassays and included two social and one solitary species: *Halictus ligatus*, *Apis mellifera* and *Ptilothrix bombiformis* (rose mallow bees), respectively. No other bee species tolerated imidacloprid as well as honey bees did, which exhibited no appreciable mortality and only modest paralysis across concentration. In contrast, native bees either lived shorter lives, experienced longer paralysis, or endured both. Overall, longevity decreased with concentration linearly for social bees and non-linearly for solitary species. The percentage of a bee's captive lifespan spent paralyzed increased logarithmically with concentration for all species, although bumble bees suffered longest. Of greatest concern was comparable debilitation of agriculturally valuable solitary bees at both low and high sublethal rates of imidacloprid.

## Introduction

As agriculture expands, so too does our reliance on insecticidal products with novel modes of action and perceived relative safety to non-target species, especially bee pollinators of

URL: https://data.nal.usda.gov/dataset/raw-data-imidacloprid-effects-native-bees.

**Funding:** This project was funded by a United States Department of Agriculture-Agricultural Research Service Cooperative Research Information System (USDA-ARS-CRIS) Project, number 6404-21430-001-00D.The funders had no role in study design, data collection and analysis, decision to publish, or preparation of the manuscript.

**Competing interests:** The authors have declared that no competing interests exist.

numerous food crops. Nevertheless, during the past 40 years, global population declines have been reported for bumble bees (*Bombus* spp.), western honey bees (*Apis mellifera* L.) and perhaps wild solitary bees [1]. Suspected causes of these bee declines and their possible interactions include natural enemy activity, degraded or lost habitat, and more controversially, exposure to persistent systemic insecticides [2–5]. The latter cause has been the most concerning due to the widespread use of systemic neonicotinoid insecticides, especially imidacloprid. Neonicotinoids, like their namesake, nicotine, bind to nicotinic acetylcholine receptors (nAChRs), hyper-stimulating the insect's central nervous system, thus causing irreversible paralysis and finally death. Consequently, neonicotinoids like imidacloprid, while relatively safe to birds and mammals, are thought to have impinged upon or jeopardized the health of numerous bee species for six main reasons. First, products made from systemic imidacloprid account for ~10% of the global insecticide market, are used on many bee-pollinated crops, and can contaminate surrounding wildflowers where foraging bees may come into physical contact with residues [2,6]. As of 2008, many U.S. crops planted with treated seed coats, particularly row crops, can account for as much as 80% of the neonicotinoid market [7]. Second, adult bees could also imbibe floral and extrafloral nectar laced with highly water-soluble imidacloprid that host plants acquire from foliar sprays or soil drenches [2]. Third, bee larvae may consume imidacloprid dust, which adult bees may collect along with pollen taken from flowers in fields containing neonicotinoid-treated seeds [8]. Fourth, the long half-life of neonicotinoids or their toxic breakdown products under low light conditions result in the contamination of soil, water and plants for 25–1155 days at insecticidal levels ranging anywhere from 5ppb to >200ppb [9–13]. Fifth, persistent neonicotinoid residues could then contaminate other pollinator nesting materials such as water, soil, mud, and leaves [14–19]. Sixth, bees undeterred or unaware of imidacloprid treated areas are prone to bioaccumulating harmful levels of the insecticide in their tissues and nests [20].

To date, assessments of pollinator sensitivity to neonicotinoids have focused on social and solitary species of manageable cavity-nesting bees [10,21–25]. An estimated 20,000 additional unmanaged bee species, their pollination services, and 35% of resulting fruit, seed, and nut yields may face continued, yet unassessed losses due to long-lasting residual toxicity of systemic pesticides [2]. Depending on the species of manageable bee, exposure to imidacloprid can be sublethal from ~1–100 ppb and fatal from 40–800 ppb [9]. Both wild and managed bees foraging from flowering food crops such as squash, pumpkin, and sunflower had the potential to ingest and carry back to their nests sublethal amounts of imidacloprid within the 5–100 ppb range [2,11,12,18]. These sublethal levels of pesticide, while not acutely fatal to domesticated bees, reportedly reduce foraging efficiency and lifespan through impaired neural function related to flight, situational awareness, navigation, learning, and memory [26–31]. Likewise, grave losses in nest productivity can occur as sublethal rates of neonicotinoids degrade one or more components of a managed bee's reproductive fitness such as longevity, virility, and fecundity [5,10,21,32–34].

For the first time, we compare the immediate health risks of a neonicotinoid, i.e., imidacloprid, on both managed honey bees and eleven species of wild native bees including agriculturally valuable pollen specialists. Risk assessments for *Apis mellifera* can differ from those for non-*Apis* bees. However, we attempted to test imidacloprid health effects on numerous pollinator species using a single standardized protocol. Probit analysis is normally used to estimate the lethal dose or concentration at which a certain proportion of an insect population is expected to die. As such, lethal concentration values assess only acute, lethal toxicity, not the more subtle and numerous sublethal effects expressed by bees both in the field and laboratory. Therefore, bee responses to imidacloprid toxicity in our alternative bioassay did not necessarily follow the sigmoidal curve typical of mortality. Based on preliminary data distributions of

our two response variables, i.e., longevity and paralysis, standard data transformations and subsequent analyses of covariance (ANCOVAs) were appropriate for assessing the chronic health effects of sublethal doses of imidacloprid ranging from 5–100 ppb. This residual range of imidacloprid has been shown to compromise honey bee health to some degree. We sought to expand risk assessments to include, in addition to honey bees, often overlooked solitary bee pollinators. Among these solitary species, we focused our attention on floral specialists, i.e., oligoleges, bees whose principal pollen hosts belong to the same plant genus, tribe, or family [35,36]. We also examined responses of social and solitary generalist bees, i.e., polyleges, that collect pollen from numerous unrelated plant genera and families [36]. The health responses of floral specialists to insecticides remain untested. Such a knowledge gap in native bee toxicology is surprising, given that their sheer abundance and more specialized foraging behaviors may render them the most effective pollinators of certain food crops such as blueberry, sunflower, okra, squash and pumpkin [12,37–39].

Under less controlled field conditions, we could never expect to account for every difference in the life history traits among our dozen pollinator species. However, on a smaller scale, we modified a honey bee feeding bioassay [40] to provide common living space for both honey bees and native bees. These bees fed from wicks, which served as communal "nectaries" for the delivery of sugar syrup amended with four concentrations of imidacloprid, i.e., 0, 5, 20, and 100 ppb. We chose to examine the health effects of imidacloprid at these four specific concentrations because similar rates were reported to harm worker bees in the genera *Apis* and *Bombus* [40–42], and under field conditions, were found in several sources of crop pollen and nectar [7,11–13,43]. We redesigned the honey bee bioassay [40] using two modifications to better house and feed multiple unrelated bee species, each differing in levels of activity, longevity, body size, and tongue length. For the first modification, we replaced gravity-fed bottles filled with 50% sucrose syrup with dark amber jars supplied with absorbent dental wicks to convey a 27% sugar solution to hungry bees. The second modification involved replacing bee mortality rate as the sole health response with two related variables whose individual observations did not require us to sacrifice groups of bees by a set time, which might result in excessive sampling of small local bee populations. The first variable was captive longevity per bee, or the number of days individuals stayed alive in the cages, which, at the level of the individual bee, is inversely related to mortality at the population level (see Fig 1). The second variable was a bee's duration of paralysis, or the percentage of its captive lifespan spent in a crippled state, i.e., % days paralyzed (see Fig 1). Crippled bees were distinguished from dead individuals based on abdominal pulsing and other subtle movements associated with respiration and feeding. Thus, with these modifications, we can now assess the immediate health risks of imidacloprid on the lifespan and mobility of members from multiple bee species or communities including, for the first time, agriculturally valuable wild pollinators.

## Materials and methods

### Bee material and basic ecology

Twelve bee species from 10 genera (*n* = 690 adult bees) were collected for the imidacloprid bioassays (Table 1). Bees were chosen for their level of sociality (solitary versus social) and degree of floral specialization (oligolecty versus polylecty). They were also abundant floral visitors or reported key pollinators of one or more of the following food crops: sunflower (*Helianthus annuus* L., and *H. mollis* Lam.), rabbiteye blueberry [*Vaccinium virgatum* (Reade)], squash and pumpkin (*Cucurbita pepo* L.), and okra [*Abelmoschus esculentus* (L.) Moench]. These crops grew in semi-natural trial gardens. Thus, pesticide pre-exposure of bees before placement in the bioassay units was minimal and any unknown active ingredients were less likely to

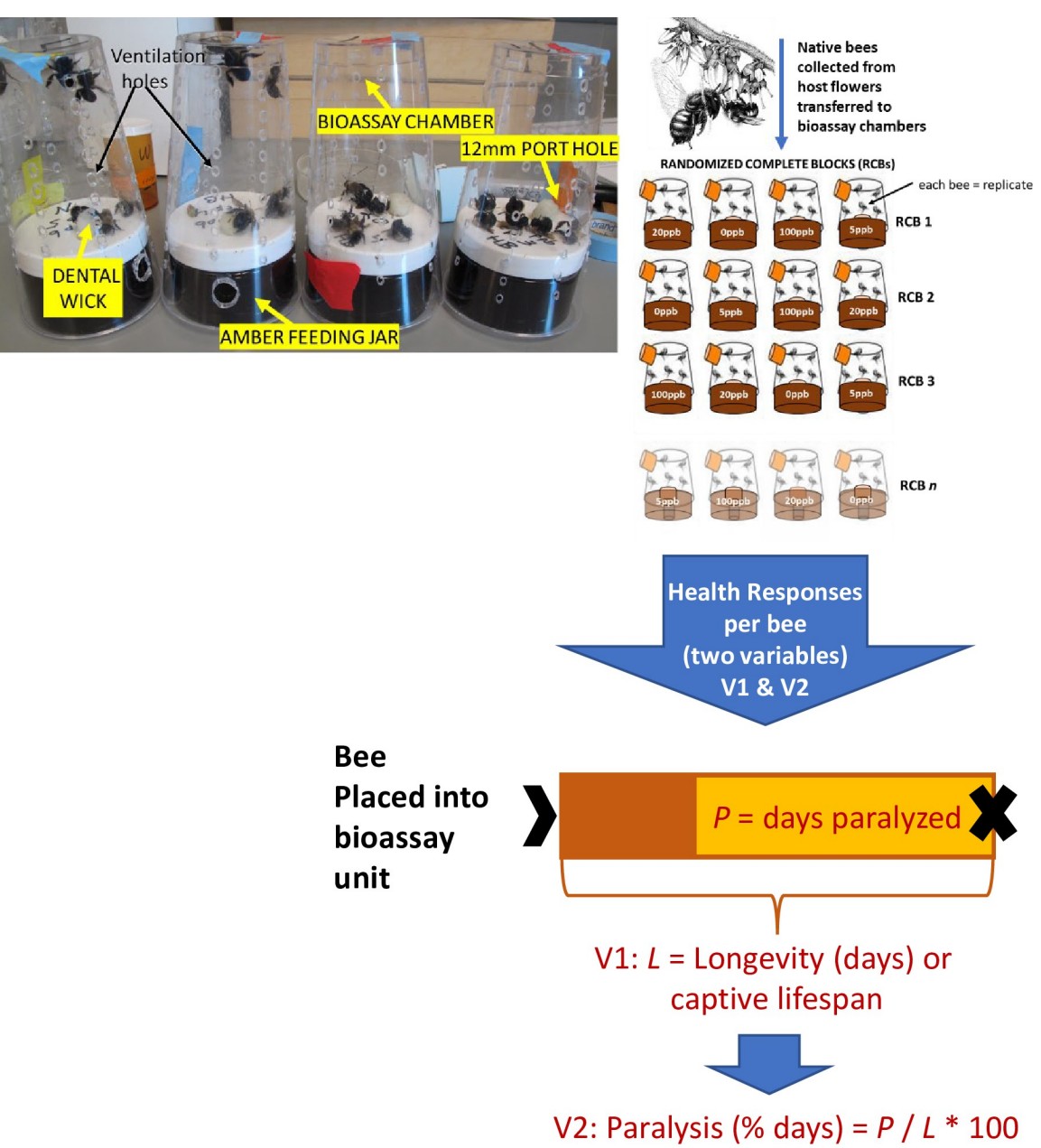

**Fig 1. Bioassay chamber construction and subsequent layout of the randomized complete block design (RCBD).** RCB is a randomized complete block consisting of one bioassay unit of each of the four imidacloprid concentrations (0 ppb or control, 5 ppb, 20 ppb and 100 ppb). Each bee in the chambers were tracked throughout its life and each represents a replicated observation or experimental unit.

bias our treatments due to the random assignment of bees to the test chambers. Workers of *Apis mellifera* L. ($n = 107$) were netted as they returned to their hive, or they were collected from nearby crop flowers between 22-Mar to 29-Dec (Table 1). Likewise, 124 bumble bees collected during a similar timeframe (22-Jun to 31-Oct) came from the same host plants as *Apis* did. For statistical purposes, health responses of locally abundant *Bombus impatiens* Cresson and *B. bimaculatus* Cresson were justifiably lumped together based on an absence of a concentration x species interaction for the genus *Bombus*. Oligolectic solitary southeastern blueberry

**Table 1. Natural history traits of the 12 bee species.** These species were in sufficient abundance for use in the imidacloprid bioassays.

| Species | Number of bee used in the bioassays | Floral hosts | Level of sociality | Pollen specialization | Collection period in south Mississippi | |
|---|---|---|---|---|---|---|
| *Apis mellifera* L. | 107 | H, OK, CP, HIVE | eusocial | polylectic | 22-Mar | 29-Dec |
| *Bombus bimaculatus* Cresson | 124 | H, OK, CP, BB | eusocial | polylectic | 23-Jun | 31-Oct |
| *Bombus impatiens* Cresson | | H, OK, CP, BB | eusocial | polylectic | 23-Jun | 31-Oct |
| *Habropoda laboriosa* (F.) | 84 | BB | solitary | oligolectic | 28-Feb | 9-Mar |
| *Halictus ligatus* (Say) | 67 | H | primitively social | polylectic | 6-Jul | 3-Oct |
| *Melissodes communis* Cresson | 32 | H, OK, CP | solitary | polylectic | 6-Jun | 29-Sep |
| *Melissodes bimaculata* (Lepeletier) | | OK, CP | solitary | polylectic | 9-Jun | 20-Jun |
| *Peponapis pruinosa* (Say) | 86 | CP | solitary | oligolectic | 5-Jun | 20-Jun |
| *Ptilothrix bombiformis* (Cresson) | 74 | OK | solitary | oligolectic | 12-Jul | 31-Jul |
| *Svastra aegis* (LaBerge) | 66 | H | solitary | oligolectic | 26-Sep | 26-Oct |
| *Trachusa zebrata* (Cresson) | 25 | H | solitary | oligolectic | 29-Sep | 26-Oct |
| *Xenoglossa strenua* (Cresson) | 25 | CP | solitary | oligolectic | 5-Jun | 20-Jun |
| H: *Helianthus annuus, H. mollis* | | | | | | |
| OK: okra, *Abelmoschus esculentus* (L.) Moench | | | | | | |
| CP: *Cucurbita pepo* L. | | | | | | |
| BB: rabbiteye blueberry, *Vaccinium virgatum* syn. *ashei* Reade | | | | | | |
| HIVE: hive bees | | | | | | |

bees, *Habropoda laboriosa* (F.), (*n* = 84) were captured between 27-Feb—10-Mar from flowering *V. virgatum* bushes. A primitively eusocial sweat bee, *Halictus ligatus* (Say), our physically smallest species, forages from a broad array of floral hosts including sunflowers [44] from which we collected 67 adults from 5-Jul to 4-Oct. Solitary polylectic long-horned bees, *Melissodes bimaculatus* (Lepeletier) and *M. communis* Cresson, visited *Hibiscus*, *Cucurbita*, and *Helianthus* from 6-Jun to 20-Sep. Like the bumble bee data, observations from both species of *Melissodes* were justifiably combined to boost sample size to 32 adults. Native squash bees, *Peponapis pruinosa* (Say) and *Xenoglossa strenua* (Cresson), are solitary ground-nesting bees and narrow oligoleges of *Cucurbita*. A total of 86 *Peponapis* and 25 *Xenoglossa* were collected from 5-Jun—20-Jun. We collected 74 rose-mallow bees, *Ptilothrix bombiformis* (Cresson) from okra between 12-Jul and 31-Jul. *Ptilothrix* are solitary ground-nesters and narrow oligoleges of Malvaceae including okra and *Hibiscus*. Narrow oligoleges of sunflowers collected from 26-Sep to 26-Oct for our bioassays included a long-horned bee, *Svastra aegis* (LaBerge) (*n* = 66), and a native megachilid, *Trachusa zebrata* (Cresson) (*n* = 25). Dichotomous keys [44,45] aided the senior author in determining species identification. Pinned vouchers reside in a private entomological collection at the USDA-ARS Thad Cochran Southern Horticultural Laboratory, Poplarville, MS, USA.

### Pollinator collecting, handling, and imidacloprid treatment

Bees were collected from flowers with a net or an aerated pill vial, chilled on ice, and within 30–60 min, were installed into bioassay cages. These cages were made from clear plastic drinking cups supplied from below with a feeding mechanism made from absorbent dental wicks (Fig 1). Cups measured 13.5 cm high and 8 cm wide at their open tops, which also served as the bottom of the cage. Each cup contained ~80 3-mm-diameter ventilation holes (Fig 1), but some cages were left undrilled to prevent the escape of small-bodied bees such as *Halictus ligatus*. A 12-mm diameter hole in the side of a cup, which was sealable with tape, served as a port

for removing dead bees (Fig 1). Inverting the cup over the opaque white lid of a dark amber feeding jar, and pressing downward, formed a snug seal that caged bees within a bioassay unit. A wick protruded through a hole in the lid of the feeding jar filled with the imidacloprid-laced syrup (Fig 1). To ensure uninterrupted syrup flow through the wicks, the concentration of the sugar in the base solution was lowered from the standard 50% used in honey bee feeders [40] to 27%. This lower percentage of sucrose is still suitable for a bee's caloric needs. Besides, plants in the field provide bees with far smaller allotments of a very dilute nectar, often a few microliters of 5–10% sugar solution per floral visit [46]. The second route of exposure was dermal, whereby bees perched themselves on feeding wicks saturated with the treated syrup.

Depending on the success of our bee captures and the docility of the pollinators caught each day, around 4–8 adults of both sexes from 1–4 different species were assigned to a bioassay unit with ~1–5 units delegated to a given imidacloprid concentration. Assigning a diversity of bees to the same bioassay cage permitted accurate tracking without unduly stressing bees with additional chilling and paint-marks. None of the bees exhibited any animosity toward one another, except for the occasional bout of assertiveness with a leg or two raised in front of an interloper (no contact or animosity displayed). There was one exception though, male *Ptilothrix bombiformis*. The frequent mauling of other subjects by male *Ptilothrix* required housing these disruptive bees in their own dedicated bioassay unit. At least 20–24 bioassay units were placed within two cardboard trays with one tray per rack sitting inside an unlit Percival E-30B growth chamber (Fig 1). The environment within a chamber set at 20°C and 65% humidity was the most amenable to our many bee species (Fig 1). The rack supporting an array of 4 cages (or block) was alternated daily to normalize any microclimatic gradient generated by a growth chamber's single internal fan. Providing no lighting inside the chambers helped to simulate the dark interior of a bee's nest, slow the photo-inactivation of imidacloprid [33], and curtail syrup fermentation. Bees remained inside the bioassay units until they died, during which time, they acquired the pesticide in two ways. First, they were orally exposed when they licked cotton dental wicks (Absorbal®) saturated with 27% (1.25 M) sucrose syrup containing one of four of the following concentrations of technical-grade water-soluble Fluka imidacloprid (Pestanal, CAS # 138261-41-3, Sigma Aldrich, St Louis, MO): 0 ppb or control, 5 ppb, 20 ppb, or 100 ppb. No solvent was needed to mix the imidacloprid in the sugar syrup. Along with the 0 ppb control (sugar syrup only), we chose these three concentrations because (1) they fell within a range of neonicotinoid residues reported in the floral rewards of known crop hosts of our chosen bee species, and (2) imidacloprid at these same or similar levels was found to diminish the reproductive success of honey bee colonies through a reduction in worker longevity and brood production [10,23,40,47,48].

To detect the onset of paralysis or sickness, as well as the time of death for each bee, we removed bioassay units daily from the growth chambers taking 15 to 30 min to assess the health of the bees inside. The two health responses defining pollinator intoxication included a bee's captive longevity and the percentage of that lifespan spent immobilized (paralyzed, Fig 1). Living bees including paralyzed subjects elicited appendicular or tegmental movements, no matter how slight. Paralyzed bees remained motionless for the most part, but sometimes moved in a sluggish fashion and were observed to evert their tongues to feed from the dental wicks. Some bees displayed the telltale signs of imidacloprid toxicity like tremors and convulsions. Tremors involved rapid twitching of legs, wings, or tarsi, whereas convulsions manifested themselves as bees bouncing around in an uncoordinated manner. A dead bee was nonresponsive even after prodding with a pointed probe. Upon death, most bees adopted a rigid species-specific death pose. For instance, dead *Halictus* curled up into a "tight ball." Legs and wings of dead *Ptilothrix* were positioned well forward and away from the thorax, giving cadavers a crab-like appearance. *Apis*, *Melissodes*, *Svastra*, *Trachusa*, *Bombus*, and *Habropoda* died

with proboscides (tongues) fully extended. A clear death pose followed by a failure of bees to recover after 48 h eliminated the possibility of us misconstruing a bee's death for death feigning or thanatosis. The second health response assayed was captive longevity or the number of days a bee stayed alive in a bioassay unit. Longevity, which is inversely related to mortality, was assessed for each of the 690 captured bees.

### Experimental design and data analyses

Arrays of four bioassay units with their respective imidacloprid concentrations were replicated in randomized complete blocks (RCBs, Fig 1). Data regarding longevity (days) were linearized before performing analyses of covariance (ANCOVA) using the $log_{10}$ transformation with all generalized linear models meeting their convergence criteria at $P \leq 0.05$. Likewise, data on duration of paralysis (percentage days paralyzed) were arcsine-square root transformed with all generalized linear models converging. Data presented in figures represent real bee responses, whereas transformed observations helped satisfy normality and heteroscedasticity assumptions of the ANCOVA models.

Influence of taxon, concentration, sex, date of capture, and their interactions on bee longevity and paralysis were analysed with three-way analyses of covariance using the GLM procedure in SAS 9.4 [49]. One-way analyses of covariance tested health responses of bee species by sex. Two-way analyses of covariance tested the effects of imidacloprid intoxication on social versus solitary bees and polyleges versus oligoleges. Longevity and paralysis were compared for pairs of bee genera by choosing a new baseline genus (control) in the 3-way ANCOVA models. $T$-values in ANCOVA tables indicated statistical differences between two mean health responses, i.e., t-values were negative when mean response $\mu_2 > \mu_1$ at $P \leq 0.05$, and positive when $\mu_1 > \mu_2$ at $P \leq 0.05$. Imidacloprid concentration (ppb) served as a continuous variable and covariate in all ANCOVA models and then was treated as a discrete (class) variable in PROC GLM, which was necessary to run Tukey's HSD tests that separated the mean bee health responses across the four concentrations (0, 5, 20, and 100 ppb). $T$-test comparisons within the ANCOVA models are given for each social versus solitary bee species. $T$-test values were nearly identical for oligoleges versus polyleges because all oligoleges were solitary and all social species were polylectic. Two species of *Melissodes* were the only solitary polyleges examined and their responses mirrored that of the other solitary bees.

### Results

The overall mean health responses of bees to imidacloprid, based on captive longevity and paralysis, varied among pollinator taxa (Table 2B–2L). In a pattern opposite that of the other bee taxa, honey bee workers tended to live longer at the two intermediate sublethal rates of imidacloprid, not shorter (i.e., 5 and 25 ppb, Table 2C, Fig 2A), although not significantly so (see Fig 2A). *Apis*, *Halictus* ($P > 0.35$), *Bombus* ($P > 0.40$), and *Ptilothrix* ($P > 0.10$) were similarly long-lived and among the longest-lived bees in our bioassays ($F_{9, 688} = 33.82$, $P < 0.0001$, Fig 2A–2D). Lifespan of *Halictus* bees was somewhat uniform at lower rates from 0–20 ppb but dropped 50% at 100 ppb (Fig 2C). All bee taxa suffered paralysis; although, *Apis* (Table 2), *Melissodes* (Fig 3B', $P > 0.20$), *Trachusa* (Fig 3D', $P > 0.60$) and *Xenoglossa* (Fig 2E', $P > 0.60$) displayed a comparable tolerance to imidacloprid-induced paralysis. The other six bee genera suffered steeper and higher rates of paralysis (Figs 2 and 3, $F_{9, 688} = 14.50$, $P < 0.0001$). Although *Bombus* and honey bees shared a similar lifespan while in captivity, the lifespan of bumble bees decreased at 20 and 100 ppb (Fig 2B) and they experienced the highest rate of paralysis beginning at 5 ppb, lasting longer than most other bees by 37 percentage points (p.p.) (Table 2D, Fig 2B'). Comparing lifespans of a sweat bee with worker honey bees, primitively

**Table 2. Summary of ANCOVA results for the effects of four imidacloprid concentrations (0, 5, 20, 100 ppb), collecting date, pollinator sex, and interactions on longevity and paralysis for 10 genera (taxa) representing 12 spp. of bees.** ANCOVA results for taxon comparisons with the baseline taxon, *Apis mellifera*, are shown in the first line for each group (i.e. 2K vs 2H, 2O vs 2H, 2S vs 2H, and so forth).

| Longevity (days) | | | | Days paralysed (%) | |
|---|---|---|---|---|---|
| | **t** | **P** | | **t** | **P** |
| A. Model | 5.52 | <0.0001 | | 7.17 | <0.0001 |
| Intercept | 5.61 | <0.0001 | | -2.88 | <0.005 |
| **B. Taxa (Genus/species)** | 5.77 | <0.0001 | | 14.52 | <0.0001 |
| C. *Apis mellifera* (baseline for inter-taxa comparisons) | — | — | | — | — |
| D. *Bombus* 2 spp. | -0.83 | 0.4073 | | 9.75 | <0.0001 |
| E. *Halictus ligatus* complex | 0.91 | 0.3623 | | 3.55 | <0.0005 |
| F. *Habropoda laboriosa* | -6.97 | <0.0001 | | 5.39 | <0.0001 |
| G. *Melissodes* 2 spp. | -7.97 | <0.0001 | | 1.20 | 0.2294 |
| H. *Peponapis pruinosa* | -10.6 | <0.0001 | | 5.64 | <0.0001 |
| I. *Xenoglossa strenua* | -2.77 | <0.01 | | 0.65 | 0.5139 |
| J. *Ptilothrix bombiformis* | -1.64 | 0.1010 | | 2.61 | <0.01 |
| K. *Svastra aegis* | -8.92 | <0.0001 | | 3.11 | <0.005 |
| L. *Trachusa zebrata* | -6.67 | <0.0001 | | -0.36 | 0.7171 |
| **J. Date of capture** | 3.12 | <0.002 | | -0.36 | 0.7216 |
| K. *Apis mellifera* | 0.87 | 0.3882 | | 5.60 | <0.05 |
| L. *Bombus* 2 spp. | -1.16 | 0.2484 | | 1.05 | 0.2946 |
| M. *Halictus ligatus* complex | 0.13 | 0.9003 | | -0.70 | 0.4876 |
| N. *Habropoda laboriosa* | -6.97 | <0.0001 | | 5.39 | <0.0001 |
| O. *Melissodes* 2 spp. | -7.97 | <0.0001 | | 1.20 | 0.2294 |
| P. *Peponapis pruinosa* | -10.6 | <0.0001 | | 5.64 | <0.0001 |
| Q. *Xenoglossa strenua* | -2.77 | <0.01 | | 0.65 | 0.5139 |
| R. *Ptilothrix bombiformis* | -1.64 | 0.1010 | | 2.61 | <0.01 |
| S. *Svastra aegis* | -8.92 | <0.0001 | | 3.11 | <0.005 |
| T. *Trachusa zebrata* | -6.67 | <0.0001 | | -0.36 | 0.7171 |
| **U. Sex (female–male)** | -0.63 | 0.5266 | | 1.03 | 0.3039 |
| V. *Apis mellifera* | —[a] | —[a] | | —[a] | —[a] |
| W. *Bombus* 2 spp. | -1.82 | 0.0697 | | -0.73 | 0.4665 |
| X. *Halictus ligatus* complex | 1.06 | 0.2909 | | -0.55 | 0.5859 |
| Y. *Habropoda laboriosa* | 1.05 | 0.2943 | | -1.71 | 0.0880 |
| Z. *Melissodes* 2 spp. | -0.55 | 0.5849 | | -1.43 | 0.1538 |
| AA. *Peponapis pruinosa* | 0.98 | 0.3296 | | -1.26 | 0.2074 |
| AB. *Xenoglossa strenua* | -1.56 | 0.1329 | | 1.84 | 0.0793 |
| AC. *Ptilothrix bombiformis* | 2.33 | <0.05 | | -1.01 | 0.3117 |
| AD. *Svastra aegis* | 1.71 | 0.0885 | | -0.54 | 0.5900 |
| AE. *Trachusa zebrata* | -0.39 | 0.6941 | | -1.23 | 0.2204 |
| **AF. Concentration (covariate)** | -3.71 | <0.001 | | 12.52 | <0.0001 |
| AG. *Apis mellifera* | -0.42 | 0.6768 | | 8.16 | <0.0001 |
| AH. *Bombus* 2 spp. | -2.28 | <0.05 | | 12.71 | <0.0001 |
| AI. *Halictus ligatus* complex | -4.14 | <0.0001 | | 4.32 | <0.0001 |
| AJ. *Habropoda laboriosa* | -2.90 | <0.005 | | 11.95 | <0.0001 |
| AK. *Melissodes* 2 spp. | -0.66 | 0.5114 | | 3.55 | <0.005 |
| AL. *Peponapis pruinosa* | -3.72 | <0.0005 | | 9.04 | <0.0001 |
| AM. *Xenoglossa strenua* | -2.06 | 0.0512 | | 3.56 | <0.005 |
| AN. *Ptilothrix bombiformis* | -2.20 | <0.05 | | 6.85 | <0.0001 |
| AO. *Svastra aegis* | -0.32 | 0.7479 | | 4.27 | <0.0001 |

*(Continued)*

**Table 2.** (Continued)

| Longevity (days) | | | | Days paralysed (%) | |
| --- | --- | --- | --- | --- | --- |
| | *t* | *P* | | *t* | *P* |
| AP. *Trachusa zebrata* | -1.34 | 0.1929 | | 2.12 | <0.05 |
| AQ. taxa *concentration | 1.77 | 0.0703 | | 5.82 | <0.0001 |
| AR. concentration*sex (F-M) | -0.15 | 0.8809 | | 0.94 | 0.3246 |

[a]only females (workers) assessed for *Apis mellifera*.

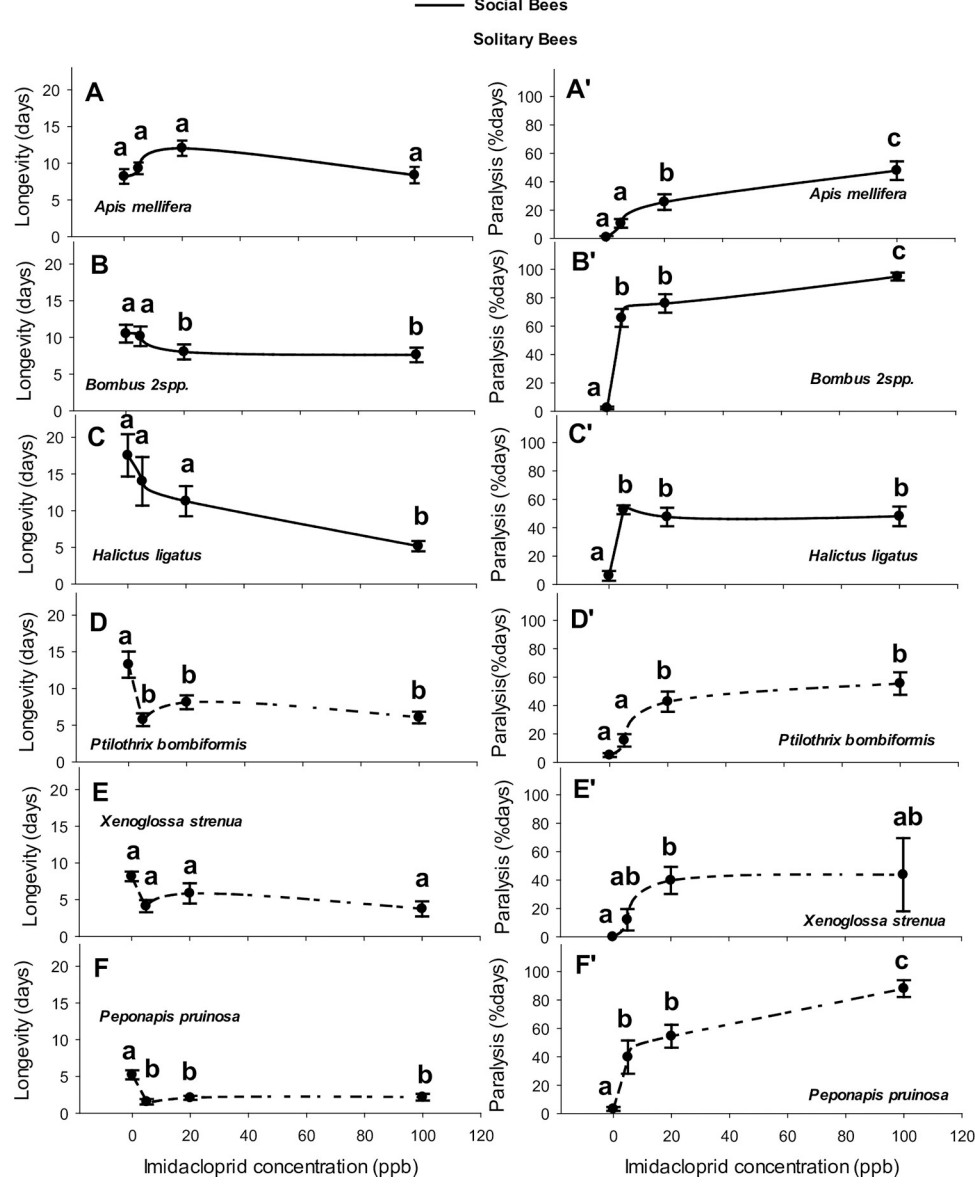

**Fig 2. Relationships between imidacloprid concentration in 27% sucrose syrup and longevity (days) and percentage of days that bees stayed paralysed (paralysis, % days) for the three social and three of the seven solitary bee genera.** Symbols and bars represent mean responses (untransformed) ± 1 SEM. Lines are shown to illustrate trends in the raw data.

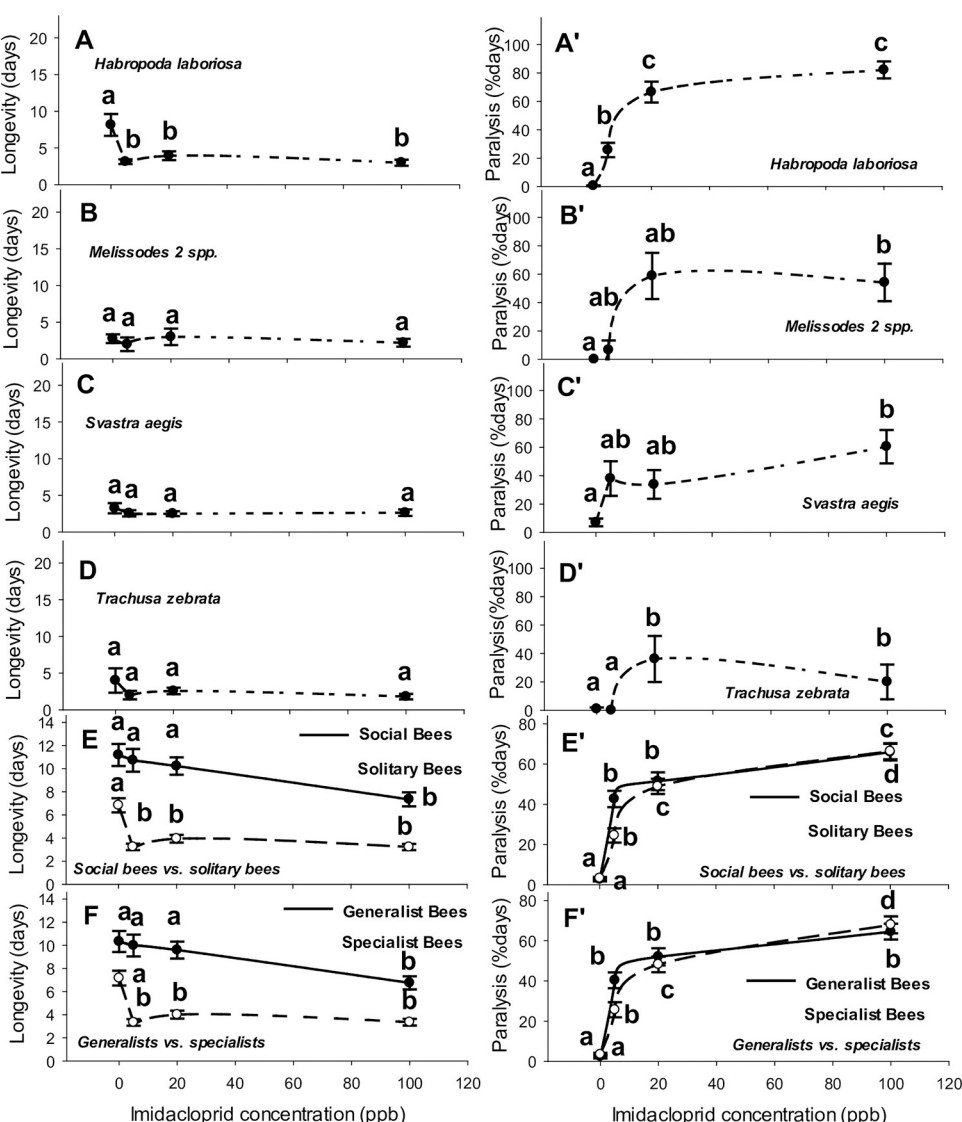

**Fig 3. Relationships between imidacloprid concentration in 27% sucrose syrup and longevity (days) and percentage of days bees remained paralysed (paralysis, % days) for the 4 other solitary bee genera as well as the combined genera of solitary bees, social bees, oligolectic bees, and polylectic bees.** Symbols and bars represent untransformed mean responses ± 1 SEM. Lines are shown to illustrate trends in the raw data.

social *H. ligatus* lived overall for the same length of time (Fig 2C), but across the four concentrations, the shapes of these bees' respective longevity curves were nearly opposite, with *H. ligatus* suffering a clear reduction in lifespan at 100 ppb, the rate that also induced temporary, non-fatal tremors and convulsions (Fig 2A and 2C). *Halictus ligatus* became immobile before honey bees did, and remained paralyzed 18 p.p. longer, whether at 5 ppb, 20 ppb or 100 ppb (Table 2E, Fig 2C'). Honey bees outlived *Habropoda laboriosa* by 5 days on average (Fig 3A) and stayed active for 21 p.p. more days (Table 2F, Fig 3A'). *Habropoda* lifespan dropped by ~50% when feeding on imidacloprid from 5–100 ppb (Fig 3A) and 30%– 80% of that time, the bees were largely immobile. *Apis* outlived *Melissodes* 2 spp. by 7 days (Fig 3), but both genera exhibited a similar duration of paralysis (22 to 32%), however adults of this solitary bee species remained more active than honey bee workers did at 0 and 5 ppb (Table 2G, Figs 2A'and 3B').

Compared with *Apis* workers, *P. pruinosa* lived 6.5 fewer days, losing 50% of its longevity at concentrations ≥5 ppb (Fig 2F) and suffering 23 p.p. longer paralysis (Table 2H, Fig 2F'). Although honey bees outlived *X. strenua* females by 3.5 days (Fig 2E), both species were comparable in their resilience to losses in longevity and durations of paralysis (22%, Table 2I, Fig 2E'). *Ptilothrix bombiformis* lived as long as *Apis* did on average across all treatments (Fig 2D), but this solitary bee species suffered a reduced lifespan at ≥20 ppb imidacloprid and experienced slightly longer paralysis (7 p.p. longer paralysis, Table 2J, Fig 2D'). In contrast to the other bees assayed, *Ptilothrix* females collected later during host anthesis outlived their males by 4 days. *Apis* workers outlived *Svastra* and *Trachusa* by ~7 days (Fig 3C and 3D; Table 2K and 2L, respectively), although imidacloprid at any rate did not alter the captive lifespans of adult bees from these three genera (Figs 2A, 3C and 3D). Mean paralysis of *Svastra* bees lasted 10 p.p. longer than it did for *Apis* (Fig 3C'), whereas paralysis for *Apis* and *Trachusa* were of equal duration, on average (17 to 22%) (Table 2K and 2L, Fig 3D').

Compared with honey bees, *Halictus*, and *Ptilothrix* were equally long-lived, whereas mean captive longevities for *Habropoda*, *Melissodes*, *Peponapis*, *Xenoglossa*, *Svastra*, *and Trachusa* were shorter. Paralysis affected all adult bees in our bioassays (Table 2AG–2AP), but *Melissodes*, *Xenoglossa* and *Trachusa* shared a duration of paralysis equivalent to that of *Apis mellifera* (Table 2G, 2I and 2L, Figs 2 and 3). Later collected bees tended to live longer in captivity (Table 2J). However, the date of capture had no appreciable effect on the captive longevities of *Apis*, *Bombus*, *Halictus*, *Melissodes*, *Xenoglossa*, *Svastra*, and *Trachusa*. Captive longevity, even in control cages, was shorter for older adults of *Habropoda* and *Peponapis* and longer for *Ptilothrix*. Duration of paralysis was lengthier for later-collected *Apis*, *Svastra*, and *Trachusa*, and unchanged for bees from the other seven genera irrespective of capture date (Table 2J–2T). Except for *P. bombiformis*, a species with shorter-lived males, both sexes of other bee taxa shared comparable longevities and paralysis (Table 2U–AE). Imidacloprid concentration did not appreciably alter the longevity of *Apis* (Table 2AG, Fig 2A), *Melissodes* (Table 2AK, Fig 3B), *Xenoglossa* (Table 2AM, Fig 2E), *Svastra* (Table 2AO, Fig 3C), or *Trachusa* (Table 2AP, Fig 3D). Concentration determined the portion of a bee's lifespan spent paralyzed (Table 2AG–2AP), but individuals from some taxa were paralyzed disproportionally longer than others at higher concentrations of imidacloprid (Table 2AQ). Health responses to imidacloprid concentration were similar for male and female bees of a given species (Table 2AR).

Along with comparisons of each native bees' health responses with those of honey bees, responses between co-foraging native species were also compared (Table 3). Of the two genera of wild blueberry pollinators tested, *Bombus* outlived *Habropoda* by 3 days, yet bumble bees exhibited 16 p.p. longer paralysis (Table 3A, Figs 2B, 2B', 3A and 3A'). Among *Cucurbita* pollinators, *Xenoglossa* outlived *Peponapis* by 2.5 days and suffered 23 p.p. fewer days of paralysis (Table 3B, Fig 2E, 2E', 2F and 2F'). *Peponapis* and *Melissodes* were similarly short-lived, ~ 2.5 days in captivity, although *Peponapis* endured 13 p.p. longer paralysis (Table 3C, Figs 2F, 2F', 3B and 3B'). *Xenoglossa* outlived *Melissodes* by 3 days, yet both spent a similar proportion of their lifespan paralyzed (22 to 32%, Table 3D, Figs 2E, 2E', 3B and 3B'). Of the wild sunflower pollinators, *Trachusa* and *Svastra* lived ~3–5 d in captivity, but *Trachusa* were paralyzed for 15 p.p. fewer days (Table 3E, Fig 3C, 3C', 3D and 3D'). *Halictus* outlived *Trachusa* by 9 days yet the sweat bee also withstood 21 p.p. longer paralysis (Table 3F, Figs 2C, 2C', 3D and 3D'). Compared with *Halictus*, *Svastra* died 9 days earlier. *Halictus* and *Svastra* were similarly paralyzed (Table 3G, Figs 2C, 2C', 3C and 3C'). *Bombus* and *Halictus* were equally long-lived, although *Bombus* exhibited 12 p.p. more days of paralysis (Table 3H, Fig 2B, 2B', 2C and 2C'). Although *Halictus* outlived *Melissodes* by 9.5 days, paralysis lasted for a similar duration in both genera (Table 3I, Figs 2C, 2C', 3B and 3B'). Captive longevity and paralysis were similar for *Trachusa*, *Svastra*, and *Melissodes* (Table 3G, 3I, and 3J, Fig 3B, 3B', 3C, 3C', 3D and 3D').

**Table 3. Summary of ANCOVA results for the effects of four imidacloprid concentrations (0, 5, 20, 100 ppb) on longevity and paralysis (days paralysed) for pairs of co-foraging species of wild bees.**

| Longevity (days) | | | | Days paralysed (%) | |
|---|---|---|---|---|---|
| | *t* | *P* | | *t* | *P* |
| A. *Habropoda* vs *Bombus* | -6.25 | <0.0001 | | -3.56 | <0.0005 |
| B. *Xenoglossa* vs *Peponapis* | 3.92 | <0.0001 | | -3.22 | <0.005 |
| C. *Peponapis* vs *Melissodes* | 0.34 | 0.7331 | | 2.78 | <0.05 |
| D. *Xenoglossa* vs *Melissodes* | 3.70 | <0.0005 | | -0.48 | 0.6308 |
| E. *Svastra* vs *Trachusa* | 0.12 | 0.9074 | | 2.46 | <0.05 |
| F. *Halictus* vs *Trachusa* | 6.76 | <0.0001 | | 2.83 | <0.005 |
| G. *Halictus* vs *Svastra* | 8.97 | <0.0001 | | 0.48 | 0.6308 |
| H. *Halictus* vs *Bombus* | 1.60 | 0.1095 | | -4.97 | <0.0001 |
| I. *Halictus* vs *Melissodes* | 8.44 | <0.0001 | | 1.39 | 0.1657 |
| J. *Svastra* vs *Melissodes* | 0.68 | 0.4984 | | 1.01 | 0.3122 |
| K. *Trachusa* vs *Melissodes* | 0.44 | 0.6587 | | -1.28 | 0.2000 |
| L. *Ptilothrix* vs *Melissodes* | 6.42 | <0.0001 | | 0.63 | 0.5311 |

*Trachusa* and *Melissodes* collected from sunflowers responded similarly to intoxication (Table 3K, Fig 3B, 3B', 3D and 3D'). Comparing wild okra pollinators, *Ptilothrix* outlived *Melissodes* by 6 days, whereas both experienced a similar duration of paralysis (Table 3L, Figs 2D, 2D', 3B and 3B'). Mean maximum paralysis was highest at 100 ppb for most of our bee species but plateaued sooner at 5 ppb for the much smaller bee, *Halictus ligatus* (Fig 2C').

Generally, generalist, social bees outlived solitary bees in the bioassays (Fig 3E and 3F) ($t = 7.04$, $P < 0.0001$). Mean longevities of adult bees feeding on imidacloprid dropped for social generalists at 100 ppb and more steeply for solitary specialists at rates ≥5 ppb (Fig 3E and 3F). The only solitary polyleges collected were two species of *Melissodes* whose collective longevity was rather short, yet the pattern of mean longevity across the four concentrations mirrored those of the other solitary bees despite individual means not differing statistically (Table 2 AK, Fig 3B). Paralysis began occurring for both social and solitary bees fed 5 ppb or more imidacloprid, but the highest paralysis rate occurred at 100 ppb (Fig 3E' and 3F'). Paralysis of captive bumble bees lasted longer than it did for any other bee (Fig 2B') resulting in a slightly longer mean duration of paralysis for social generalists ($t = 2.11$, $P < 0.05$; Fig 3E' and 3F'), even after adding in the more resilient honey bees (Fig 2A') and *Halictus* (Fig 2C') to this group.

## Discussion

The focus of risk assessments regarding imidacloprid and similar compounds is their toxicity to managed cavity-nesting pollinators such as honey bees, bumble bees, and orchard mason bees. Health responses to environmental toxins have received almost no clinical evaluation for wild native bees that nest below ground, which are perhaps more important than honey bees as pollinators of certain plants and related food crops, e.g., blueberry [38], squash [39] and okra [48]. However, our inability to move or shield native bees from pesticide residues is believed to leave them vulnerable to the toxicity of persistent systemic insecticides like neonicotinoids [17].

Pesticide risk assessments involving one or a few bee species often begin with small-scale, well-controlled, and low-cost cage studies. Soon after, these preliminary trials transition to larger-scale and logistically complicated open field tests. Even in small-scale laboratory bioassays, assessing the ecotoxicological risks of pesticides to native bees is a logistical challenge. In

particular, a more stressful environment often associated with bioassays could amplify the harmful effects of a pesticide beyond that of a wild field bee experiencing a more normal nesting context. Therefore, we minimized stress on bees by reducing handling time and forgoing the customary chilling and paint-marking prior to installation. Instead, we monitored the health responses of one or both sexes of readily identifiable species of social and solitary bees within the same bioassay unit. We chose for our bioassay to quantify longevity and paralysis as health responses instead of percentage mortality. Mortality is the preferred health response tracked in most pollinator risk assessments involving abundant, managed bees. However, waiting for a proportion of bees to die on a set day to obtain a single observation of percentage mortality might entail a scale of replication that, while fine for a managed bee species, could exhaust local populations of native bees. Thus, a single bee rather than a cohort serves as an experimental replicate. To further help alleviate stress on bees, we simulated realistic exposure to imidacloprid by using saturated dental wicks to dose bees in two ways. First, through oral exposure whereby adult bees imbibed sucrose solution from an absorbent dental wick. This feeding mechanism better simulated the uptake of nectar from flowers. Also, a bee was not required to climb up to a suspended feeder or fly toward it, which were necessary for honey bees to do in the previous bioassay [40]. Second, bees could experience dermal exposure as they walked over or perched on feeding wicks, behaviors akin to those of field bees that might otherwise walk, rest, and groom on contaminated plant surfaces.

Overall, sensitivity to imidacloprid based on longevity and paralysis depended on one or more bee life history traits such as taxa, sex, capture date, level of sociality and degree of floral specialization. Bee paralysis lasted longer at higher concentrations for all captured bees, but individuals from certain species were more resilient. In contrast, captive longevities were shortened for such high value crop pollinators as specialist solitary bees. Vulnerability of smaller-bodied bees to imidacloprid intoxication is believed to be rather high [17]. Although we did not measure or weigh every bee placed into our bioassays, our results indicate variation in adult body size for our 12 species poorly predicted sensitivity to imidacloprid. Small-bodied halictids seem quite resilient to sublethal doses, although *H. ligatus* was the only bee species in our bioassays that exhibited the classic, albeit transient, symptoms of neurotoxicity, i.e., tremors and confusion, and only at 100 ppb. Instead, body size may better predict the chance that a bee will encounter floral patches contaminated with varying quantities of imidacloprid. Smaller-bodied bees with shorter flight ranges may face sporadic, but acute dangers, if they chose to nest in cropland habitats that receive direct pesticide inputs [17]. Larger-bodied bees while foraging for food may face a more predictable, yet chronic risk of exposure, should they encounter patches of insecticidal residues along their longer flight paths [50]. Even if bees receive a non-fatal dose of imidacloprid, our bioassays show that most will still suffer paralysis. Unlike our laboratory bees, a paralyzed bee in the field, like any debilitated bee, would likely succumb to predation, starvation, or some other peril. In the larger-bodied, non-halictine bees, tremors, if any, are not visible at sublethal concentrations and paralysis took longer to manifest. A delayed onset of bee paralysis may result from chronic exposure to low doses of imidacloprid, which may be slow to bind to nicotinic receptors [51–53].

Female bees often dwarf males in size, but both sexes, for the most part, show equal resilience or sensitivity to imidacloprid neurotoxicity. In fact, the smallest males to the largest females of non-*Apis* bees were similarly long-lived, except *Ptilothrix bombiformis*, a species whose females outlived males by ~4 days. We began caging captive *Ptilothrix* males by themselves as soon as we observed them antagonizing their fellow cage mates, a similar territorial behavior they also direct toward other bees at host blooms in the field [54]. Thus, we suspect pollinator lifespan may decrease further due to heightened stress resulting from imidacloprid intoxication punctuated with periods of inter-bee animosity. Moreover, adult males of

protandrous solitary bees and older female bees may face the more adverse effects of imidacloprid. Later collected *Habropoda* and *Peponapis* died earlier in treatment and control units, whereas later-collected *Apis*, *Svastra* and *Trachusa* became paralyzed sooner. These findings coupled with a weak link between a species' specific body size and overall health responses to imidacloprid infer other factors like inter-bee interactions may better predict tolerance.

Bee longevity differed quantitatively, decreasing linearly with imidacloprid concentration in social bees. Yet, many of our solitary bee species suffered a far more precipitous drop from 0 to 100 ppb. Additional factors may also determine a bee's sensitivity to imidacloprid or its capacity to mitigate harmful side-effects. A bee's ability to detoxify imidacloprid or at least tolerate toxicity somehow was linked to its species level of sociality and floral specialization. Social polyleges lived the longest while feeding on all three imidacloprid concentrations, outliving both specialist and generalist solitary bees. The only solitary polylectic bees collected belonged to two species of *Melissodes*, which like other solitary bees, were quite sensitive to imidacloprid intoxication. Although *Melissodes*, *Svastra*, and *Trachusa* were the shortest-lived bees, e.g., ~3–5 days, all three species exhibited the same quadratic responses in longevity as did the other four genera of solitary bees. Essentially, imidacloprid at 5 ppb, 20 ppb or 100 ppb reduced the lifespans of solitary bees with a slight tick upwards at 20 ppb. A consistent quadratic pattern in the longevity of native bees indicates that higher imidacloprid concentrations cause bees to ingest smaller lifetime doses of the pesticide. Perhaps at higher concentrations, digestive muscles become paralyzed sooner while gut tissues suffer damage, thereby reducing the feeding rate and hence the uptake of the poison. Such peristaltic paralysis is known to affect honey bees feeding on imidacloprid [40]. Whether this phenomenon affected the health of our eight species of solitary bees remains unknown because we chose not to monitor syrup uptake for fear that the periodic disassembly of the feeding jars would introduce another stressor into the bioassay. Nevertheless, our results corroborate those of Gregorc et al. [40], which now confirm that even trace amounts of imidacloprid can debilitate honey bees, but native bees can suffer even more.

Our bioassays and field trials by others confirm a honey bee's tolerance for trace quantities of imidacloprid in their food. Our results further corroborate a much higher sensitivity to imidacloprid in bumble bees when compared with honey bees [22,55]. Nevertheless, acute population declines in eusocial species like honey bees may occur with widespread colony failure after the poisoning of queens. Chronic losses in nest productivity may further amplify these losses due to mass mortality among worker bees in established colonies [17]. In a stingless bee, *Scaptotrigona postica*, workers began to die within 24 to 48 hours after feeding on imidacloprid concentrations identical to ours [48]. Thus, imidacloprid residues as low as 5–100 ppb can pose a high risk to eusocial bee species in both tropical and temperate climates. Likewise, among temperate species of solitary bees that we tested, specialist bees such as *H. laboriosa*, *S. aegis*, *Ptilothrix bombiformis*, and *P. pruinosa* lived shorter lives or became immobile at concentrations as low as ~5 ppb or 20 ppb imidacloprid; field rates comparable to those measured elsewhere in the floral rewards of these bees' principal cultivated hosts: blueberry, sunflower, okra and squash, respectively. These findings do not necessarily implicate imidacloprid uptake by bees as a major cause of global colony collapse and population decline in honey bees. Rather, a higher sensitivity to imidacloprid toxicity shown by bumble bees and some solitary bee species raises the concern that losses in wild bee diversity, while unseen by farmers, will become more obvious as a reduction in pollination service and then crop yield [22]. However, among the dozen native bees tested, three specialist crop pollinators tolerated low doses of imidacloprid rather well. They included *Xenoglossa strenua*, a squash and pumpkin pollinator, *Trachusa zebrata*, a sunflower pollinator, and to some extent *Ptilothrix bombiformis*, a pollinator of okra, *Hibiscus* and cotton. These three solitary oligoleges along with social *Apis* and

*Halictus* may already possess some adaptation for dealing with both neonicotinoid molecules and similarly toxic plant compounds at the molecular level, including species-specific mutations in detoxification genes, varying protein-binding affinities, and changes to a toxin's skeletal structure [56–59].

Despite a bee's chemically complicated environment filled with interacting agrichemicals including pesticides, adjuvants, plant protectants, and growth regulators, risk assessments are often restricted to a single active ingredient or insect species due to daunting logistics and budgetary constraints. In reality, adult bees while foraging are exposed to a myriad of active ingredients from the crop and elsewhere [43] that could interact with each other as well as with imidacloprid, which might affect pollinator sensitivity in the field and laboratory assessments. In our bioassay, however, wild bees and any compounds they may carry were randomly assigned to all treatments including the control, and hence the degree of pre-exposure to pesticides should be normalized across treatment. In the case of honey bees, however, acaricides are routinely applied inside colonies and at high enough concentration could prolong worker lifespan at lower imidacloprid densities due to perhaps a reduced uptake of contaminated honey [40] (i.e., hormesis-like effect). Despite the diverse chemical space traversed by honey bees, it is their availability to researchers that has made *Apis mellifera* the pollinator of choice in most pesticide risk assessments and, as such, these bees are generally regarded as a suitable proxy for other bee species. Yet, data herein show this assumption to be somewhat invalid. In fact, the observed species-specific sensitivities to imidacloprid in this study indicates no single bee taxon could serve as a proxy for the thousands of other pollinator species with varying life histories [60]. Although, the environmental persistence of a popular and systemic active ingredient like imidacloprid in an agricultural insecticide warrants a more in-depth risk assessment to non-target species such as bees. Realistic testing and later modelling of pollinator risk to any common insecticide should incorporate smaller-scale bioassays involving numerous bee species along with one or perhaps more interacting co-factors such as food type, other pesticide residues, biome, floral host, life history (including body size, age, or growth stage), metabolic chemistry, and genetics [4,17,34,61]. Through bioassays of species' sensitivities to pesticides, we can begin to model and more accurately predict the demographic changes the pesticides can have on whole pollinator communities.

## Conclusions

By examining the sublethal effects of imidacloprid on a dozen species of common North American bees, we confirmed the resilience of honey bees and the extreme vulnerability of bumble bees to this insecticide [25]. We also expanded the list of vulnerable bee species to include oligolectic pollinators of some valuable fruit and vegetable crops. Because most of the solitary bees assayed were floral specialists with a heightened sensitivity to imidacloprid under captive conditions, polylectic social species, as a group, outlived solitary oligoleges by 2–6 days, on average. Low imidacloprid concentrations from 5 ppb to 100 ppb shortened life expectancy for both social and solitary bees, more dramatically in the latter. Imidacloprid-induced declines in longevity trended linearly with concentration for social species and curvilinearly for solitary bees, indicating that both low and high sublethal rates can weaken the health of some wild pollinators [9,52]. Although imidacloprid-induced paralysis lasted the longest for bumble bees and the shortest for *Apis*, *Trachusa* and *Halictus*, paralysis for all 10 bee genera followed a similar lognormal dose-response. Under laboratory conditions, bee body size only weakly predicted sensitivity to imidacloprid-induced paralysis and mortality. In contrast, other life history traits such as pollinator sex, relative age, level of sociality and degree of floral host specialization had a much stronger bearing on the sensitivity of individual bees to

imidacloprid. Thus, it is inadvisable during a pollinator risk assessment with a bioassay, to base imidacloprid-induced health responses of untested species of solitary and other social bees on a highly eusocial proxy like the worker honey bee. If our bioassay results hold true in the chemical-rich environment of a commercial crop, the uptake of even trace amounts of a persistent neurotoxic insecticide like imidacloprid (e.g., 5 ppb) could impinge on the mobility, survival, and resulting pollination services of many species of wild bees, especially abundant solitary oligoleges.

## Acknowledgments

We thank Gene Blythe, Scott Langlois, and Ray Morris for access to their gardens, and Mississippi Power for access to large wild sunflower stands. Appreciation to Katherine Parys for her advice on some bee identifications. Thanks to Eric Stafne and Gene Blythe for their early review of the manuscript. The use of trade, firm, or corporation names in this publication is for the information and convenience of the reader. Such use does not constitute an official endorsement or approval by the U.S. Department of Agriculture (USDA) or the Agricultural Research Service, or Mississippi State University of any product or service to the exclusion of others that may be suitable.

## Author Contributions

**Conceptualization:** Blair Sampson, Aleš Gregorc, John Adamczyk.

**Data curation:** Blair Sampson, Aleš Gregorc, Christopher Werle.

**Formal analysis:** Blair Sampson, Aleš Gregorc, Christopher Werle.

**Funding acquisition:** Blair Sampson, Aleš Gregorc, John Adamczyk, Patricia Knight.

**Investigation:** Blair Sampson, Aleš Gregorc, Christopher Werle, John Adamczyk.

**Methodology:** Blair Sampson, Aleš Gregorc, Christopher Werle.

**Project administration:** Blair Sampson, Aleš Gregorc, John Adamczyk.

**Resources:** Blair Sampson, Aleš Gregorc, Christopher Werle, Patricia Knight.

**Software:** Blair Sampson, Aleš Gregorc, Christopher Werle, Patricia Knight.

**Supervision:** Blair Sampson, Aleš Gregorc, Christopher Werle, John Adamczyk, Patricia Knight.

**Validation:** Blair Sampson, Aleš Gregorc, Mohamed Alburaki, Christopher Werle, John Adamczyk.

**Visualization:** Blair Sampson, Aleš Gregorc, Christopher Werle, John Adamczyk.

**Writing – original draft:** Blair Sampson, Aleš Gregorc, Mohamed Alburaki, Christopher Werle, Shahid Karim, John Adamczyk, Patricia Knight.

**Writing – review & editing:** Blair Sampson, Aleš Gregorc, Mohamed Alburaki, Christopher Werle, Shahid Karim, John Adamczyk, Patricia Knight.

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
