## [Decision Letter · Decision Letter 0]

31 Jan 2023

PONE-D-22-33169Sensitivity to imidacloprid insecticide varies among some social and solitary bee species of agricultural valuePLOS ONE

Dear Dr. Sampson,

Thank you for submitting your manuscript to PLOS ONE. After careful consideration, we feel that it has merit but does not fully meet PLOS ONE’s publication criteria as it currently stands. Therefore, we invite you to submit a revised version of the manuscript that addresses the points raised during the review process.

We look forward to receiving your revised manuscript.

Kind regards,

Adam G Dolezal

Academic Editor

PLOS ONE

Journal Requirements:

   "We thank Gene Blythe, Scott Langlois, and Ray Morris for access to their gardens, and Mississippi Power for access to large wild sunflower stands. Appreciation to Katherine Parys for her advice on some bee identifications. Thanks to Eric Stafne and Gene Blythe for their early review of the manuscript. This project was funded by a United States Department of Agriculture548 Agricultural Research Service Cooperative Research Information System (USDA-ARS-CRIS) Project, number 6404-21430-001-00D. The use of trade, firm, or corporation names in this publication is for the information and convenience of the reader. Such use does not constitute an official endorsement or approval by the U.S. Department of Agriculture (USDA) or the Agricultural Research Service, or Mississippi State University of any product or service to the exclusion of others that may be suitable."

  "This project was funded by a United States Department of Agriculture-Agricultural Research Service Cooperative Research Information System (USDA-ARS-CRIS) Project, number 6404-21430-001-00D.The funders had no role in study design, data collection and analysis, decision to publish, or preparation of the manuscript. "

Additional Editor Comments:

Please find the comments from two reviewers below. Overall, they are quite positive and I think you can easily address their comments.

I think Rev 1's comment about figure resolution may have been a problem in the submission documents, but try to make sure your revision figures are as high resolution as possible.

An issue both reviewers brought up was the potential for other agrochemicals in the environment in which the bees were collected could affect your results, and both pose some interesting points about this issue. I suggest considering their comments on this topic. I also advise seriously considering the point Rev. 2 raises about the conflation of collection date with bee age.

Reviewers' comments:

Reviewer's Responses to Questions

**Comments to the Author**

1. Is the manuscript technically sound, and do the data support the conclusions?

Reviewer #1: Yes

Reviewer #2: Yes

2. Has the statistical analysis been performed appropriately and rigorously? 

Reviewer #1: Yes

Reviewer #2: Yes

3. Have the authors made all data underlying the findings in their manuscript fully available?

Reviewer #1: Yes

Reviewer #2: Yes

4. Is the manuscript presented in an intelligible fashion and written in standard English?

Reviewer #1: Yes

Reviewer #2: Yes

5. Review Comments to the Author

Reviewer #1: Research summary

Sampson et al. assessed the immediate health risk of the neonicotinoid insecticide imidacloprid in 12 different bee species with varying levels of socialization, floral specialization, and body size collected from several crops and fed them with sublethal doses of imidacloprid. It was found that imidacloprid has a higher effect on the longevity and paralysis of native bees than Apis mellifera and that the sensitivity of imidacloprid varies significantly among different bee species.

The findings of this study are important to highlight the difference in sublethal effects of the insecticide on Apis mellifera, which is used as a proxy for wild bees, with solitary and other social native bees suffering from the effects of pesticides. Although having an essential role in pollination, these bees have fewer toxicity and risk assessment studies.

The manuscript is well-written, the methods and data presentation are adequate, and the results support the conclusions. There are minor revisions and comments presented below:

1. The imidacloprid concentrations described in the introduction (line 124) could benefit from more recent references (for example, Graham et al., DOI: 10.1038/s41598-021-96249-z for blueberry fields in Michigan). In the discussion (lines 483 - 487), there is a statement of “field rates comparable to those measured elsewhere in the floral rewards of these bees’ principal cultivated hosts: blueberry, sunflower, okra, and squash, respectively” of 5 to 100 ppb, but it does not present references that support all of these crops.

2. There is information that would be important to add in the methods section: Which solvent was used to dilute the imidacloprid stock solution before adding it to the sucrose solution? Was the relative volume of imidacloprid stock solution high in the different concentration groups? Was there a vehicle group to eliminate the effect of this solvent on longevity?

3. There must be spaces between the numerical values and unit symbols throughout the manuscript (especially concentrations).

4. All figures could benefit from better resolution. It is not easy seeing all components of the chamber in Fig1 and the components of the graphs in Figures 2 and 3. When the figures are zoomed in, it becomes pixelated. Also, in Fig3, it could be better for visualization to separate the solitary bee genera from the combined genera of all bees.

5. The collected bees are adult individuals and already were exposed to other chemicals that could inflict interactions with imidacloprid that could influence the results. Discussing these interactions in light of the different collection locations and crops and pesticide spraying season could be interesting; for example, a higher concentration of an antagonist near a honey bee hive could explain the opposite pattern of honey bees living longer on lower imidacloprid concentration. Is there a possibility of hormesis-like mechanisms of resistance?

Reviewer #2: Sampson et al.’s study, “Sensitivity to imidacloprid insecticide varies among some social and solitary bee species of agricultural value,” provides much-needed insight into the potential ecological and agricultural risks that neonicotinoid contamination on pollinator attractive crops can have across a variety of bee species. Over a two-year period, 690 bees from 12 different species were collected on agricultural crops and subjected to neonicotinoid bioassays at several sublethal, but field realistic levels of imidacloprid. The authors found that imidacloprid exposure had a negative, linear effect on social bees and a non-linear negative effect on solitary species. Furthermore, polylectic species had an increased longevity compared to oligolectic counterparts. Introducing controls with toxicological assays with native bees is particularly difficult, and while some comments below point out potential methodological shortcomings (not at the fault of the authors), results from this study are crucial, as they point out a fundamental flaw with how risk assessment strategies are currently operating; assuming that managed species such as honey and bumble bees can represent all species. The manuscript is well written and will be a worthy and much needed addition to the broader pollinator toxicological literature, however the following comments should be addressed prior to publication.

Major comments:

- The authors appear to conflate the age of bees with the date of capture, “the relative ages of bees based on the date of capture” (line 322). As bee phenologies and life spans vary significantly across the species sampled, date of capture and age of bee should not be used interchangeably. Furthermore, age of and individual can have a significant effect on its sensitivity to various insecticides (Rinkevich et al. 2015). Without any age-related analyses such as wing wear, this lack of actual bee age should be discussed. Honey bee bioassays are typically conducted with uniform age groups. While this is almost impossible for most solitary species as they respond poorly to lab settings, attempting to address age via wing wear or discussing potential shortcomings is needed to qualify these findings.

- The authors collected sampled bees off agricultural plants. This could introduce a base level of neonicotinoid exposure that likely is not uniform across samples due to date of capture and differential systematic activity and uptake of neonicotinoids in plant tissues. Even if the crops on which bees were collected were not treated, the span of collection times could coincide with planting dates for seed treated crops, which could introduce a drift contamination component. For bees collected in the field (particularly native bees) it is almost impossible to escape neonicotinoid contamination in agricultural environments.

Minor comments:

Introduction:

- Lines 66-69: While 10% market share is relevant to imidacloprid, it undersells the scale and breadth of neonicotinoid use, particularly in agriculture. This is particularly true for crops that utilize seed coatings where, as of 2008, neonicotinoids make up 80% of the market (Klingelhöfer et al. 2022); market share has undoubtedly increased.

Methods:

- Lines 202-204: “Assigning a diversity of bees to the same bioassay cage permitted accurate tracking without unduly stressing bees with additional chilling and paint-marks.” While there likely was a reduction in stress as a result of omitting chilling and marking periods, stress could have been non-uniformly introduced by placing different species with varying levels of aggression in the same cage. This set-up provides an opportunity for aggressive behavior rooted in the phenomena of interference competition which is seen in field settings between honey and native bees (Paini 2004). While the stress of bioassays is brought up in the discussion, there is no mention of potential stress caused by close interspecific proximity.

Results:

- Lines 270-273: If results are not significant, then a directional pattern of longevity should not be asserted as a potential trend.

6. PLOS authors have the option to publish the peer review history of their article (what does this mean?). If published, this will include your full peer review and any attached files.

Reviewer #1: No

Reviewer #2: No

---

## [Author Response · Author response to Decision Letter 0]

13 Apr 2023

 "We thank Gene Blythe, Scott Langlois, and Ray Morris for access to their gardens, and Mississippi Power for access to large wild sunflower stands. Appreciation to Katherine Parys for her advice on some bee identifications. Thanks to Eric Stafne and Gene Blythe for their early review of the manuscript. This project was funded by a United States Department of Agriculture548 Agricultural Research Service Cooperative Research Information System (USDA-ARS-CRIS) Project, number 6404-21430-001-00D. The use of trade, firm, or corporation names in this publication is for the information and convenience of the reader. Such use does not constitute an official endorsement or approval by the U.S. Department of Agriculture (USDA) or the Agricultural Research Service, or Mississippi State University of any product or service to the exclusion of others that may be suitable."

 "This project was funded by a United States Department of Agriculture-Agricultural Research Service Cooperative Research Information System (USDA-ARS-CRIS) Project, number 6404-21430-001-00D.The funders had no role in study design, data collection and analysis, decision to publish, or preparation of the manuscript. " Response: Funding text has been removed from the acknowledgement and the funding information declares is correct and does not require updating. Thanks.

Additional Editor Comments:

Please find the comments from two reviewers below. Overall, they are quite positive and I think you can easily address their comments.

I think Rev 1's comment about figure resolution may have been a problem in the submission documents, but try to make sure your revision figures are as high resolution as possible.

An issue both reviewers brought up was the potential for other agrochemicals in the environment in which the bees were collected could affect your results, and both pose some interesting points about this issue. I suggest considering their comments on this topic. I also advise seriously considering the point Rev. 2 raises about the conflation of collection date with bee age.

Reviewers' comments:

Reviewer's Responses to Questions

Comments to the Author

1. Is the manuscript technically sound, and do the data support the conclusions?

Reviewer #1: Yes

Reviewer #2: Yes

2. Has the statistical analysis been performed appropriately and rigorously? 

Reviewer #1: Yes

Reviewer #2: Yes

3. Have the authors made all data underlying the findings in their manuscript fully available?

Reviewer #1: Yes

Reviewer #2: Yes

4. Is the manuscript presented in an intelligible fashion and written in standard English?

Reviewer #1: Yes

Reviewer #2: Yes

5. Review Comments to the Author

Reviewer #1: Research summary

Sampson et al. assessed the immediate health risk of the neonicotinoid insecticide imidacloprid in 12 different bee species with varying levels of socialization, floral specialization, and body size collected from several crops and fed them with sublethal doses of imidacloprid. It was found that imidacloprid has a higher effect on the longevity and paralysis of native bees than Apis mellifera and that the sensitivity of imidacloprid varies significantly among different bee species.

The findings of this study are important to highlight the difference in sublethal effects of the insecticide on Apis mellifera, which is used as a proxy for wild bees, with solitary and other social native bees suffering from the effects of pesticides. Although having an essential role in pollination, these bees have fewer toxicity and risk assessment studies.

The manuscript is well-written, the methods and data presentation are adequate, and the results support the conclusions. There are minor revisions and comments presented below:

1. The imidacloprid concentrations described in the introduction (line 124) could benefit from more recent references (for example, Graham et al., DOI: 10.1038/s41598-021-96249-z for blueberry fields in Michigan). In the discussion (lines 483 - 487), there is a statement of “field rates comparable to those measured elsewhere in the floral rewards of these bees’ principal cultivated hosts: blueberry, sunflower, okra, and squash, respectively” of 5 to 100 ppb, but it does not present references that support all of these crops. Response: Graham et al. 2021 is a pivotal reference and we incorporated it throughout the manuscript, many thanks for the suggestion. We also added addition supporting references to fortify this claim. 

2. There is information that would be important to add in the methods section: Which solvent was used to dilute the imidacloprid stock solution before adding it to the sucrose solution? Was the relative volume of imidacloprid stock solution high in the different concentration groups? Was there a vehicle group to eliminate the effect of this solvent on longevity? Response; imidacloprid is highly soluble water, therefore it was dissolved within the sugar syrup quite readily. We make this more clear in the text. Many thanks. The volume of the syrup remained the same in jars at the different concentrations. The control group was fed just sugar water. Thanks. This has been clarified in the text. 

3. There must be spaces between the numerical values and unit symbols throughout the manuscript (especially concentrations). Response: Many thanks, this has been corrected. 

4. All figures could benefit from better resolution. It is not easy seeing all components of the chamber in Fig1 and the components of the graphs in Figures 2 and 3. When the figures are zoomed in, it becomes pixelated. Also, in Fig3, it could be better for visualization to separate the solitary bee genera from the combined genera of all bees. Response: So sorry for the poor resolution, the pdf proof looked clear on my computer screen, but it printed out pixelated. I am so sorry that you had to look over such a messy output. I was hoping that it was my printer that was glitching out on me. I will be sure that the next upload will be at the highest possible resolution permitted. Many thanks.

5. The collected bees are adult individuals and already were exposed to other chemicals that could inflict interactions with imidacloprid that could influence the results. Discussing these interactions in light of the different collection locations and crops and pesticide spraying season could be interesting; for example, a higher concentration of an antagonist near a honey bee hive could explain the opposite pattern of honey bees living longer on lower imidacloprid concentration. Is there a possibility of hormesis-like mechanisms of resistance? Response: This is a great point, which we had alluded too, but you had more eloquently described. We did include our response in the last paragraph of the discussion. Many thanks. 

Reviewer #2: Sampson et al.’s study, “Sensitivity to imidacloprid insecticide varies among some social and solitary bee species of agricultural value,” provides much-needed insight into the potential ecological and agricultural risks that neonicotinoid contamination on pollinator attractive crops can have across a variety of bee species. Over a two-year period, 690 bees from 12 different species were collected on agricultural crops and subjected to neonicotinoid bioassays at several sublethal, but field realistic levels of imidacloprid. The authors found that imidacloprid exposure had a negative, linear effect on social bees and a non-linear negative effect on solitary species. Furthermore, polylectic species had an increased longevity compared to oligolectic counterparts. Introducing controls with toxicological assays with native bees is particularly difficult, and while some comments below point out potential methodological shortcomings (not at the fault of the authors), results from this study are crucial, as they point out a fundamental flaw with how risk assessment strategies are currently operating; assuming that managed species such as honey and bumble bees can represent all species. The manuscript is well written and will be a worthy and much needed addition to the broader pollinator toxicological literature, however the following comments should be addressed prior to publication. 

Major comments:

- The authors appear to conflate the age of bees with the date of capture, “the relative ages of bees based on the date of capture” (line 322). As bee phenologies and life spans vary significantly across the species sampled, date of capture and age of bee should not be used interchangeably. Furthermore, age of and individual can have a significant effect on its sensitivity to various insecticides (Rinkevich et al. 2015). Without any age-related analyses such as wing wear, this lack of actual bee age should be discussed. Honey bee bioassays are typically conducted with uniform age groups. While this is almost impossible for most solitary species as they respond poorly to lab settings, attempting to address age via wing wear or discussing potential shortcomings is needed to qualify these findings. Response: We at first wanted to assess wing wear by counting notches and gouges, devising an index to assess the level of wing wear prior to introduction. It became too stressful to keep bees immobile (through chilling) too long prior to placement into the bioassay chambers. Assessing wing wear post-mortem was possible, but alas was not performed. We just wanted to see if date of capture had an influence on bee responses to imidacloprid intoxication. We, however, did mention that date may serve as a crude measure of age. This conflation is now eliminated, and we just mention these results in terms of date of capture with no connection to the physical or chronological age of the bee. Thanks for pointing out this error on our part. Much appreciated.

- The authors collected sampled bees off agricultural plants. This could introduce a base level of neonicotinoid exposure that likely is not uniform across samples due to date of capture and differential systematic activity and uptake of neonicotinoids in plant tissues. Even if the crops on which bees were collected were not treated, the span of collection times could coincide with planting dates for seed treated crops, which could introduce a drift contamination component. For bees collected in the field (particularly native bees) it is almost impossible to escape neonicotinoid contamination in agricultural environments. Response: Thanks for the excellent recommendation. We were aware of the chance of our bees coming into contact with pesticides within and from outside of our collection sites. We tried to minimize this likelihood by gathering wild bees from crops growing in nearby trial gardens managed by our staff, which receive minimal insecticide inputs. We realize this isn’t a perfect solution, but we had hoped that including a control (0ppb imidacloprid) would account for any health effects of some unknown active ingredient brough into the chambers by the captured bees. Of course, if interactions occurred between these chemicals and imidacloprid, we could not really be capable of assessing their overall impact. But clearly, imidacloprid effects were consistent across treatments and replicate bees. We included in our results section, the following statement: “These crops grew in semi-natural trial gardens. Thus, pesticide pre-exposure of bees before placement in the bioassay units was minimal and any unknown active ingredients were less likely to bias our treatments due to the random assignment of bees to the test chambers”.

Minor comments:

Introduction:

- Lines 66-69: While 10% market share is relevant to imidacloprid, it undersells the scale and breadth of neonicotinoid use, particularly in agriculture. This is particularly true for crops that utilize seed coatings where, as of 2008, neonicotinoids make up 80% of the market (Klingelhöfer et al. 2022); market share has undoubtedly increased. Response: Thanks for the undated reference. We were focussing on imidacloprid and its uses as a spray, drench, or seed treatment. We wholeheartedly agree that your statistic is equally jaw-dropping. Thus, we included it in the introduction as follows: “. As of 2008, U.S. crops planted with treated seed coats account for as much as 80% of the neonicotinoid market (Klingelhöfer et al. 2022).” As of 2008, many U.S. crops planted with treated seed coats, particularly row crops, can account for as much as 80% of the neonicotinoid market (Klingelhöfer et al. 2022).

Methods:

- Lines 202-204: “Assigning a diversity of bees to the same bioassay cage permitted accurate tracking without unduly stressing bees with additional chilling and paint-marks.” While there likely was a reduction in stress as a result of omitting chilling and marking periods, stress could have been non-uniformly introduced by placing different species with varying levels of aggression in the same cage. This set-up provides an opportunity for aggressive behavior rooted in the phenomena of interference competition which is seen in field settings between honey and native bees (Paini 2004). While the stress of bioassays is brought up in the discussion, there is no mention of potential stress caused by close interspecific proximity. Response: We carefully monitored bees daily and all bees including honey bees showed no animosity, just occasional avoidance behavior. However, as mentioned in the text, we did observe consistent aggressive behavior from Ptilothrix males, which we isolated as soon as we noticed this behavior. Animosity was also observed by Ptilothrix males in on open field situation, so it is not a behavior that is an artifact of crowded captive conditions. Thanks for this concern and certainly it is something one might have to be aware of when caging mixed pollinator species. We added the following statement to our material and methods section: “None of the bees exhibited any animosity toward one another, except for the occasional bout of assertiveness with a leg or two raised in front of an interloper (no contact or animosity displayed). There was one exception though, male Ptilothrix bombiformis.”.

Results:

- Lines 270-273: If results are not significant, then a directional pattern of longevity should not be asserted as a potential trend. Response: Thanks so much. A poor choice of words. Reads now as “Lifespan of Halictus bees was uniform at lower rates from 0 – 20 ppb but dropped 50% at 100 ppb (Figure 2C). 

6. PLOS authors have the option to publish the peer review history of their article (what does this mean?). If published, this will include your full peer review and any attached files.

Do you want your identity to be public for this peer review? For information about this choice, including consent withdrawal, please see our Privacy Policy.

Reviewer #1: No

Reviewer #2: No

---

## [Editor Report · Decision Letter 1]

18 Apr 2023

Sensitivity to imidacloprid insecticide varies among some social and solitary bee species of agricultural value

PONE-D-22-33169R1

Dear Dr. Sampson,

We’re pleased to inform you that your manuscript has been judged scientifically suitable for publication and will be formally accepted for publication once it meets all outstanding technical requirements.

Kind regards,

Adam G Dolezal

Academic Editor

PLOS ONE

Additional Editor Comments (optional):

Thank you for your prompt resubmission. I feel that authors have done a great job responding to the reviewer comments, which has resulted in an improved manuscript. As such, I do not see a need to return this to the reviewers and recommend we accept the manuscript. I do recommend that the authors pay special attention to spelling, grammar, etc. during production to make sure they take the opportunity to fix any small typographical problems that may have been missed.
---

## [Editor Report · Acceptance letter]

24 Apr 2023

PONE-D-22-33169R1 

Sensitivity to imidacloprid insecticide varies among some social and solitary bee species of agricultural value 

Dear Dr. Sampson:

I'm pleased to inform you that your manuscript has been deemed suitable for publication in PLOS ONE. Congratulations! Your manuscript is now with our production department. 

Kind regards, 

on behalf of

Dr. Adam G Dolezal 

Academic Editor

PLOS ONE